# TimeRM: Multi-Expert Residual Modeling for Long-Term Time Series Forecasting

## Abstract

In time series forecasting, residuals capture unmodeled dynamics that are entangled with various temporal patterns, resulting in complex structures that challenge the model's predictive accuracy. To address this, we propose TimeRM, a simple yet effective residual modeling framework. We decompose the time series into periodic and residual components. The Multi-Level Decoupling (MLD) module processes the residuals, extracting sub-residuals that capture hierarchical temporal patterns and supply the main model with previously neglected dynamics. We further introduce the Multi-level Sub-residual Compensation (MSRC) module, based on the Mixture-of-Experts (MoE) architecture, which captures entangled temporal patterns, quantifies their contribution to the sequence, compensates the primary residual, and thereby reduces the impact of unmodeled dynamics. Additionally, to resolve the issue of insufficient representation of expert models when handling data with complex channel characteristics, we propose the Multi-Expert Coupling (MEC) module to predict sub-residuals, enhancing the prediction of neglected channel features through a multi-level coupling approach to improve model performance. Extensive experiments demonstrate that our TimeRM framework outperforms current state-of-the-art methods.

## 1 Introduction

Time series forecasting Kong et al. (2025); Li et al. (2024) is widely applied in domains such as finance and the Internet of Things (IoT). By analyzing and mining historical data to forecast future trends, thereby supporting applications including economic decision-making and risk management Wang et al. (2024b). Real-world time series exhibit a complex interleaving of diverse temporal patterns. Currently methods struggle to comprehensively and accurately capture random fluctuations, leading to significant deviations between predictions and actual values. This inherent complexity of mixed temporal dynamics Jin et al. (2024); Shi et al. (2025); Zhong et al. (2025) severely limits the interpretability of the models.

Decomposing a time series into sub-sequences with distinct characteristics, such as trend, seasonality, and residuals, can effectively reduce the complexity of forecasting tasks Bandara et al. (2025), enable more accurate modeling of individual dynamic patterns, and enhance the interpretability of prediction results. AutoFormer Wu et al. (2021) advances beyond conventional preprocessing-based decomposition methods by integrating a decomposition module directly into the model architecture. However, its fixed decomposition mechanism often fails to adapt to highly complex or non-stationary temporal dynamics, resulting in inaccurate separation of trend and seasonal components. CycleNet Lin et al. (2024) addresses cyclic pattern modeling through a learnable periodic parameter matrix and improves interpretability via residual filtering and spectral analysis. Nevertheless, it overlooks the informative content embedded in noise and non-stationary residuals, limiting its robustness under significant domain shifts. S2IP-LLM Pan et al. (2024) leverages STL decomposition to decouple temporal patterns and aligns them semantically with the latent space of large language models (LLMs), enabling dynamic pattern disentanglement and refined forecasting. Yet, its effectiveness hinges on the coverage and alignment fidelity of pre-trained LLM knowledge, particularly for non-stationary behaviors Ye et al. (2024b); Liu et al. (2022b). By neglecting explicit modeling of non-stationary components, current decomposition strategies constrain both the adaptability and interpretability of forecasting models.

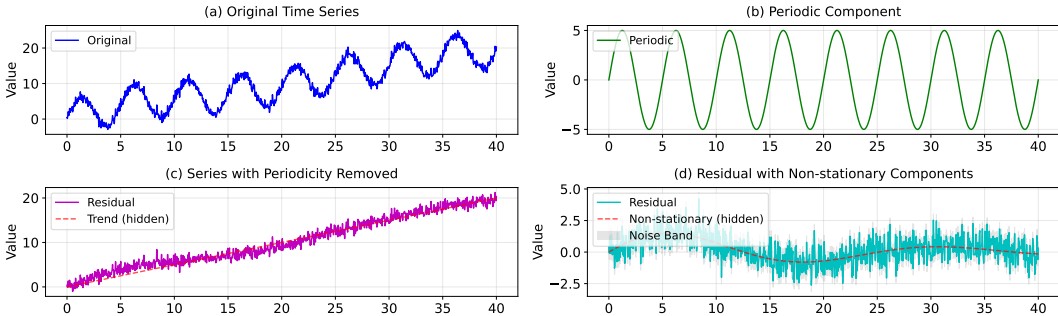

Figure 1: (a) Orign time series. (b) Periodic components in the original sequence. (c) The residual components remaining after removing the periodic components contain trend. (d) Intertwined non-stationary information within the residual components

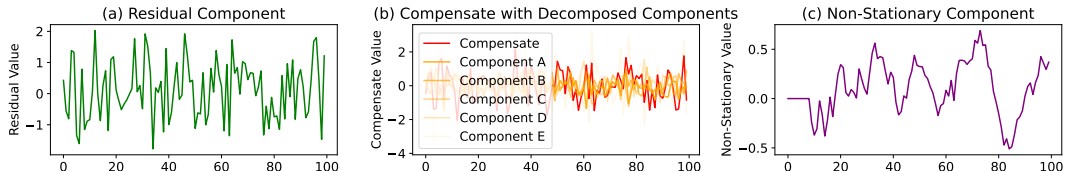

Figure 2: (a) The residual components of time series are significant in data analysis. (b) The MSRC module provides multi-compensation. (c) The residual and compensatory components are added together, and the reconstructed residual component obtained transforms the original random perturbations into statistical properties, thus making the sequence interpretable.

Recently, TimeKAN Huang et al. (2025) decouples multi-frequency components in time series by decomposing the original sequence into independent frequency bands. This approach effectively addresses the issue of uneven information density when modeling mixed-frequency signals and demonstrates improved capability in handling non-stationary behavior. TimeMixer++ Wang et al. (2025b) proposes a multi-scale, multi-resolution framework with joint time and frequency domain fusion, enabling comprehensive understanding and efficient extraction of complex temporal patterns. Time-Bridge Liu et al. (2024a) introduces an innovative architecture designed to tackle non-stationarity in long-term forecasting. It balances the removal of short-term non-stationarity to prevent spurious regression with the preservation of long-term non-stationary trends to capture global dependencies, leading to significant improvements in prediction accuracy. Despite these advancements, real-world time series often exhibit dynamically evolving patterns due to inherent non-stationarity. The decomposition and hybrid modeling strategies used in these methods are typically based on fixed or predefined assumptions, which limits their ability to adapt to sudden or gradual changes in temporal dynamics. As shown in Fig. 1, the residual components of time series contain substantial random fluctuations and reflect complex, underexplored structures within non-stationary information. These residuals may also harbor valuable latent temporal patterns that have not been fully exploited. Therefore, accurately identifying such hidden dynamics and effectively integrating multiple key features remain critical challenges for improving forecasting performance and model robustness.

This paper proposes a framework for modeling residual components. Specifically, we propose a Multi-Level Decoupling (**MLD**) module that gradually separates stable trends from residual components, allowing the multi-sub-residual to encompass time patterns varying degrees, providing complex patterns that are missing from the main residual. Subsequently, a Multi-Level Residual Compensation (**MSRC**) based on Mixture-of-Experts (MoE) strategy to capture intertwined temporal patterns along the sequence's time dimension. It quantifies the impact of the missing complex patterns on the main residual components, transforming the existing random fluctuations into quantifiable statistical characteristics. This process are shown in Fig. 2. Furthermore, the expert models within the MSRC lack channel interaction, making it difficult to analyze complex channel relationships in high-dimensional data. We have designed a Multi-Expert Coupling (**MEC**) module. This module predicts multiple potential temporal patterns at each time step, optimizes the neglected channel features, and enhances the model's predictive capability.

Our contributions can be summarized as follows:

- We propose the TimeRM algorithm for residual modeling. The residual patterns in time series are decoupled into multiple sub-residual components to finely model the underlying missing patterns, quantifying the impact of these missing patterns on the overall series.

- The MSRC module based on the MoE strategy, disentangles intertwined temporal patterns from multi-sub-residuals. It mitigates the effects of random perturbations by efficiently analyzing and compensating for missing information in the principal residual mode.

- The MEC module, designed for high-dimensional feature data, captures multiple latent temporal patterns at each time step by leveraging channel interaction. It effectively recovers previously overlooked channel features and enhances the model's predictive capability.

- Experimental results demonstrate that our method outperforms current state-of-the-art (SOTA) methods.

## 2 RELATED WORKS

### 2.1 TIME SERIES FORECASTING

The deep learning approaches of time series forecasting have gained significant attention, such as CNN Zhang et al. (2025a); Xie et al. (2025), RNN Cinar et al. (2018); Jia et al. (2024), MLP Zeng et al. (2023), and Transformer Ren et al. (2024). The Transformer's advantage in handling long-range dependencies enables it to capture long-term dependencies in stable time series. In this way, PatchTST Nie et al. (2022) enhances local dependencies by segmenting time series patches. iTransformer Liu et al. (2024b) independently embeds the entire time series of each variable as variable tokens, allowing the attention mechanism to directly focus on inter-variable correlations rather than temporal dependencies. CSFormer Wang et al. (2025a) can separately extract sequence-specific and channel-specific information while sharing parameters to promote synergy and mutual reinforcement between sequences and channels. However, the aforementioned studies exhibit limitations in modeling complex temporal patterns, such as multi-seasonality and change points Bandara et al. (2025). In this work, we decompose the time series across multiple scales and employ separate linear models for each case, effectively addressing this representational limitation.

### 2.2 TIME SERIES DECOMPOSITE

Real-world time series often exhibit multiple intertwined temporal patterns. STL decomposition Ouyang et al. (2023) separates a series into trend, seasonality, and residual components but falters in multi-seasonal settings. Bandara et al. Bandara et al. (2025) extended STL using locally weighted regression to handle multiple seasonalities. S2IP-LLM Pan et al. (2024) aligns STL-derived components with the latent space of large language models (LLMs), improving prediction of complex patterns. Some approaches adopt implicit decomposition: AutoFormer Wu et al. (2021) uses embedding layers to capture trends, TimesNet Wu et al. (2022) applies Fast Fourier Transform Wu et al. (2025); Yang et al. (2024) to identify dominant frequencies, and TimeMixer Wang et al. (2024a) downsamples data across seasonal and trend scales. However, these methods often neglect nonlinear interactions among components, leading models to favor prominent patterns while overlooking subtler dynamics. To address this, we propose a multi-level decoupling framework based on residual learning, designed to uncover hidden information in residuals and enhance model interpretability.

### 2.3 NON-STATIONARY EXPLORE

Real-world time series are typically non-stationary. Classical methods such as ARIMA Zhang (2003) enforce stationarity via differencing, but this direct transformation can degrade model performance. RevIN Kim et al. (2021) applies instance normalization to inputs and outputs separately, reducing intra-sequence variability. Arik et al. Arik et al. (2022) improve generalization by training models to forecast masked historical segments. Liu et al. Liu et al. (2023) leverage modern Koopman theory to capture underlying time-varying dynamics, offering a principled approach to

non-stationarity. Wu et al. Zhang et al. (2025b) combine Fourier and Laplace transforms in FLD-Mamba Zhang et al. (2025b) to jointly model global periodicity and local transient changes, enabling comprehensive analysis of complex non-stationary behavior. Nevertheless, existing methods struggle to effectively capture the intricate residual components carrying critical non-stationary signals. To address this, we propose a compensation strategy that decomposes residuals and quantifies the influence of their temporal patterns on the overall sequence. By learning adaptive compensation weights to characterize stochastic fluctuations, our method transforms erratic noise into interpretable statistical features, thereby enhancing the modeling and understanding of non-stationary dynamics.

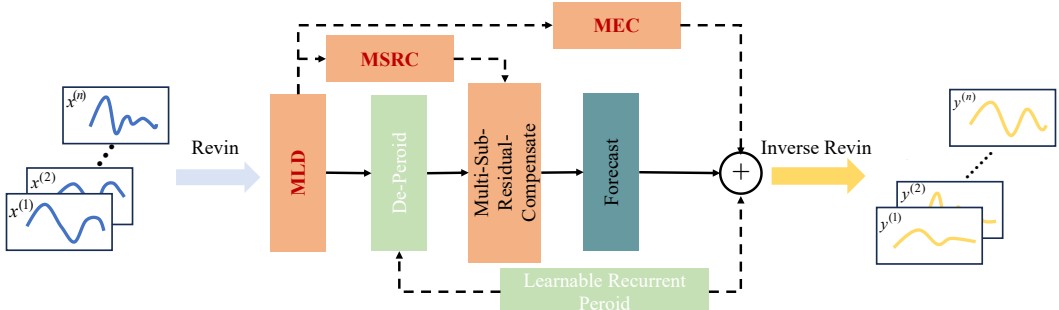

Figure 3: The overview of TimeRM

## 3 METHODS

### 3.1 OVERVIEW OF TIMERM

The overall architecture of **TimeRM** is illustrated in Fig. 3. The **MLD** module decomposes the residual into multiple sub-residuals using parallel filters, while the **MSRC** module employs a Mixture of Experts (MoE) strategy to capture diverse complex temporal patterns as a complement to the main residual. To overcome the representation limitations of MoE under high-dimensional channels, we propose the Multi-Expert Coupling (**MEC**) module, which enhances forecasting accuracy by predicting on sub-residuals and recovering missing temporal patterns, adapting to dominant trends in the data.

### 3.2 TIMERM FRAMEWORK

Given a historical multivariate time series input $x \in \mathbb{R}^{L \times C}$, the aim is to predict the future output $y \in \mathbb{R}^{N \times C}$ , where L, N is the look-back window length and the future window length, and C represents the number of variates.

**Instance Normalization.** Distributional shifts in real-world data often impair model generalization. Although existing methods such as RevIN Kim et al. (2021) and FAN Ye et al. (2024a) aim to alleviate this issue, we adopt a streamlined instance normalization (IN) approach that effectively mitigates performance degradation caused by distributional shifts, with negligible computational overhead. The method can be formulated as:

$$x_t = \frac{x - \mu}{\sqrt{\sigma^2}}, \quad y = y_t \times \sqrt{\sigma^2} + \mu, \tag{1}$$

where $x_t \in \mathbb{R}^{B \times L \times C}$ and $y_t \in \mathbb{R}^{B \times N \times C}$ denote the normalized input and denormalized output, respectively. Here, $\mu$ and $\sigma$ represent the mean and standard deviation computed over the temporal dimension.

**Multi-Level Residual Decompose.** According to the Seasonal-Trend decomposition using Loess (STL), a time series can be decomposed into three components: trend, seasonal, and residual. The seasonal component captures periodic patterns inherent in the data. To further refine the residual component, we apply a low-pass filter along the temporal dimension to extract its relatively stationary trend. By subtracting this filtered output from the original series, we obtain the high-frequency residual, as formalized below:

$$H = x - \text{AvgPool}(x). \tag{2}$$

Learning all patterns within a sequence $x$ simultaneously, including long-term trends, periodic behavior, and abrupt anomalies, often results in insufficient representation of subtle features. This is due to the model's tendency to overfit dominant patterns while underlearning weaker but meaningful signals. To address this limitation, we employ a hierarchical decomposition strategy based on successive sub-residuals (a detailed exposition is provided in Appendix A.1). This method progressively removes slowly varying, near-stationary trends through iterative filtering, generating sequences that highlight increasingly fine-grained fluctuations. In doing so, it reveals dynamic characteristics that are typically missing or poorly captured in the primary residual component.

---

**Algorithm 1** Multi-Level Sub-Residual Compensate

---

1: **procedure** MULTI-LEVEL SUB-RESIDUAL COMPENSAT($r_1, \ldots, r_n$)
2:     $z \leftarrow r$, $p = Top_k(f(z))$, where $f$ is a linear layer,         ▷ k: Expert Number
3:     **for** $i \leftarrow$ **to** $n$ **do**         ▷ Decomposition level
4:         **for** $j \leftarrow 1$ **to** $k$ **do**
5:             $c_i = Model_j(z_i[p_j])$,    where $z \in \mathbb{R}^{(B \times L) \times C}$
6:         **end for**
7:     **end for**
8:     **return** $c_1, \ldots, c_n$         ▷ Output
9: **end procedure**

---

**Multi-Sub-Residuals Compensate.** The multi-sub-residual components and the main residual component share multiple common patterns, which exhibit intrinsic correlations. Conventional approaches Zeng et al. (2023); Wu et al. (2021); Wang et al. (2025b) typically employ separate models to independently extract key features from each component before subsequent processing. This paradigm can be formulated as:

$$c_1, c_2, \ldots, c_n = \text{Model}_1(r_1), \text{Model}_2(r_2), \ldots, \text{Model}_n(r_n), \tag{3}$$

where each residual component $r_i$ is processed by a dedicated model to obtain its corresponding feature representation $c_i$. However, such methods overlook the interdependencies among temporal patterns across different components, making it difficult to accurately capture temporal characteristics and thus compromising modeling efficiency. Sub-residual components may simultaneously contain diverse temporal patterns. Due to the heterogeneous characteristics of these components, a single processing method struggles to accurately capture the embedded patterns. Inspired by the Mixture of Experts (MoE) strategy, we construct and learn a selection mechanism along the sequence dimension to identify the specific temporal patterns present at each time step. This process is shown in Algorithm 1.

Through this approach, we can effectively capture common patterns across different sub-residual components. Integrating these patterns with the main residual component allows us to quantify the impact of these missing patterns on the principal component, thereby establishing a compensation mechanism that significantly enhances the interpretability of the residual components. On the other hand, the temporal pattern $x$ exhibits inherent periodic structures; independent analysis of such periodic patterns can significantly enhance the accuracy of time series forecasting Lin et al. (2024). We employ learnable parameter matrices to remove them. It can be formulated as:

$$x_{dc} = x - C_x, \tag{4}$$

where $C_x \in \mathbb{R}^{B \times L \times C}$ denotes periodic information, the detailed introduction are provided in Appendix A.2. Subsequently $x_{dc}$ through step-by-step compensation, enhancing the Interpretability of Residual Components. It can be formulated as:

$$x_{non} = x_{dc} + c_1 + \ldots + c_n + \Delta_x, \tag{5}$$

where $x_{dc} \in \mathbb{R}^{B \times L \times C}$, $\Delta_x$ serves as an additional compensation. Considering that excessive decomposition resulting in a large number of sub-residuals can lead to the emergence of redundant information, while a small number of sub-residuals cannot provide the main model with sufficient missing pattern information, we add a bias term to address this issue.

---

**Algorithm 2** Multi-Expert Coupling

---

1: **procedure** MULTI-LEVEL-DECOMPOSITION($x$)
2:    $residuals$                                                                  ▷ Multi-Sub-Residuals
3:    $K \leftarrow [\,]$
4:    **for** $i \leftarrow 1$ **to** $n$ **do**
5:        $k \leftarrow Model(residuals[i])$
6:        **append** $k$ **to** $K$
7:    **end for**
8:    **return** $K$                                                                ▷ Output
9: **end procedure**

---

**Multi-Expert Coupling.** For multivariate time series forecasting, some datasets feature high dimensionality and exhibit complex inter-channel relationships within the data. When confronted with such challenges, a large number of expert models are required to work in collaboration. This not only increases the computational load of the models but also raises the likelihood of redundant information. To address this issue, we have devised a second compensation strategy to replaces MSRC module that involves directly predicting sub-residuals obtained from multiple decomposition methods. We view high-dimensional feature channels as the coupling of multiple temporal patterns, this approach enables the learning of useful representations and cross-channel dependencies from high-dimensional, thereby providing channel relationship compensation to the predictions made by the main model that adapts to certain dominant patterns and refining the overlooked channel interactions. This process are shown in Algorithm 2.

**Main Residual Analysis.** For the sequential characteristics in residual modeling, a lightweight Multi-Layer Perceptron (MLP) is employed to model the temporal dependencies within the sequences. Previous studies have confirmed the robustness of MLP in extracting temporal features. The MLP in TimeRM consists of two fully connected layers, with activation functions interspersed between the layers, and it is defined as follows:

$$\overline{x} = Linear(\sigma(Linear(x_{non}))), \tag{6}$$

where $x_{non} \in \mathbb{R}^{B \times L \times C}$,,$\sigma$ is the activation function. Finally, the predicted results undergo further mapping to align and fuse with the periodic information $C_y$, thereby obtaining accurate prediction results. It can be formulated as:

$$y = Linear(Dropout(\overline{x})) + C_y, \tag{7}$$

where $C_y \in \mathbb{R}^{B \times N \times C}$, $\overline{x} \in \mathbb{R}^{B \times O \times C}$. For scenarios with large channel dimension, we integrate the output of the MEC module during the prediction phase to compensate for the forecasting results. The formula is shown as follows:

$$\overline{y} = y + k_1 + \ldots + k_n + \Delta_y, \tag{8}$$

Here, $\Delta_y$ serves the same purpose as $\Delta_x$ in Eq. 5.

## 4   EXPERIMENT

### 4.1   COMPARATIVE RESULTS

TimeRM is compared with current state-of-the-art (SOTA) methods on widely used public datasets, as shown in Table 1. The results show that TimeRM ranks among the top two across all 18 metrics, demonstrating overall SOTA performance and confirming its effectiveness. Notably, the proposed MSRC strategy performs well on low-dimensional feature datasets, while MEC excels on high-dimensional ones. Complete results and analyses are provided in Appendix B.3. Overall, despite its simple architecture, TimeRM effectively enhances the model's ability to capture complex residual patterns. In the following sections, we analyze how MSRC and MEC facilitate interpretable temporal pattern mining. (Details of baselines and datasets are in Appendix B.1 and Appendix B.2.)

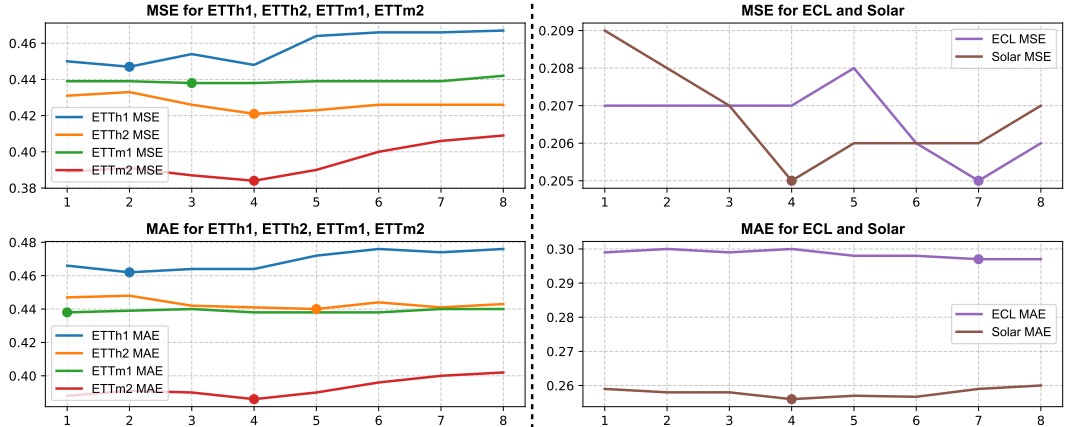

Figure 4: Impact of the number of decoupling layers on the performance of the MSRC and MEC modules, with all results assessed at a 720 step forecast horizon.

Table 1: Long-term series forecasting results for a sequence length of 96. The forecasting horizons are set to $\{96, 192, 336, 720\}$, and all results are averaged. The best and the second-best results are highlighted in **red** and **blue**, respectively. Full results are provided in Table 4.

| Models | TimeRM | | TQNet 2025 | | TimeBridge 2025 | | FLDMamba 2025 | | TimeMixer++ 2025 | | TimeMixer 2024 | | CycleNet 2024 | | iTransformer 2024 | |
|---|---|---|---|---|---|---|---|---|---|---|---|---|---|---|---|---|
| Metrics | MSE | MAE | MSE | MAE | MSE | MAE | MSE | MAE | MSE | MAE | MSE | MAE | MSE | MAE | MSE | MAE |
| ETTh1 | **0.420** | **0.433** | 0.441 | 0.434 | 0.530 | 0.507 | 0.434 | 0.430 | 0.467 | 0.450 | 0.457 | 0.441 | 0.454 | 0.448 | 0.453 | 0.453 |
| ETTh2 | **0.370** | **0.399** | 0.378 | 0.402 | 0.376 | 0.399 | 0.372 | 0.396 | 0.380 | 0.404 | 0.388 | 0.409 | 0.383 | 0.407 | 0.413 | 0.427 |
| ETTm1 | **0.371** | **0.395** | 0.377 | 0.393 | 0.452 | 0.438 | 0.389 | 0.399 | 0.384 | 0.399 | 0.379 | 0.396 | 0.407 | 0.410 | 0.400 | 0.412 |
| ETTm2 | **0.269** | **0.315** | 0.277 | 0.323 | 0.280 | 0.321 | 0.279 | 0.322 | 0.287 | 0.329 | 0.274 | 0.322 | 0.288 | 0.332 | 0.289 | 0.330 |
| Weather | **0.241** | **0.271** | 0.242 | 0.269 | 0.255 | 0.272 | —— | —— | 0.250 | 0.275 | 0.243 | 0.271 | 0.258 | 0.278 | 0.249 | 0.278 |
| Electricity | **0.165** | **0.259** | 0.167 | 0.262 | 0.206 | 0.282 | 0.170 | 0.238 | 0.186 | 0.272 | 0.168 | 0.259 | 0.178 | 0.270 | 0.194 | 0.301 |
| Solar | **0.194** | **0.254** | 0.198 | 0.256 | 0.222 | 0.254 | 0.235 | 0.256 | 0.232 | 0.273 | 0.237 | 0.302 | 0.210 | 0.261 | 0.233 | 0.262 |
| PEMS04 | **0.099** | **0.209** | 0.129 | 0.246 | —— | —— | 0.099 | 0.209 | 0.152 | 0.261 | 0.105 | 0.259 | 0.118 | 0.226 | 0.113 | 0.222 |

## 4.2 ABLATION STUDY

**Quantitative and Qualitative Analysis.** The MLD model decomposes multiple sub-residuals, and the number of decoupled layers significantly influences model performance. We conduct a quantitative analysis by varying the number of decoupled layers while keeping the number of expert models fixed in the MSRC module. To the left of the dashed line in Fig. 4, results from the MSRC module using the ETT dataset show that excessive decoupling may introduce redundant temporal patterns, impairing model accuracy. To the right of the dashed line are the results for the MEC module, where the number of expert models is set equal to the number of decoupled layers. The experiments show that the optimal number of decoupled layers is 7 for the Electricity (ECL) dataset and 4 for the Solar dataset.

Based on the optimal number of decoupling layers identified for the MSRC module, we performed a qualitative analysis of the number of expert models in the MoE decomposition strategy. The results are presented in Fig. 5, where the horizontal axis represents the number of expert models, and the lowest point on the curve indicates the optimal configuration. In our experiments, we focused on the ETTh1 dataset. Specifically, with 20 expert models, we unfolded the time series along the temporal dimension and analyzed the assignment distribution across experts. The resulting heatmap reveals one dominant expert and three other experts with relatively high activation frequencies. This observation aligns with the trend shown in the line graph, which demonstrates suboptimal performance with a single expert and peak performance with four experts. This indicates that our method effectively identifies four distinct and important temporal patterns. Furthermore, we observe that an excessive number of experts introduces redundancy, thereby degrading overall model performance.

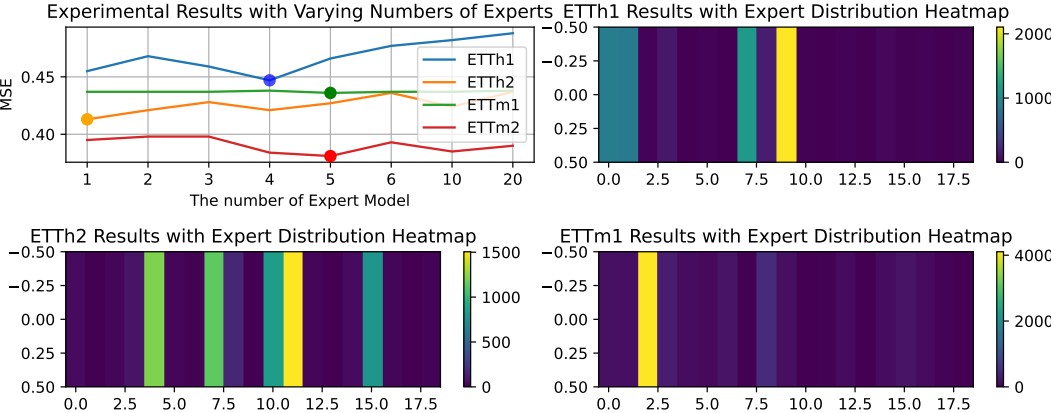

Figure 5: This study examines how the number of expert models affects system performance and analyzes the distribution of these models across various temporal patterns.

In Figure 6, we analyze the behavior of the MEC model over time on the Solar dataset. Specifically, we extract the contribution ratio of each expert's output at every time step and visualize its dynamic evolution using line plots. Furthermore, by aggregating the output features of all experts, we construct a cross-channel cross-correlation matrix and present its structure in the form of a heatmap. Notably, the regions of high correlation in the heatmap are not concentrated near the diagonal but are broadly distributed across the matrix. This indicates that the MEC module facilitates extensive interactions among experts along the channel dimension at different time steps, effectively capturing latent cross-channel temporal dependencies within the multi-branch residual architecture. This mechanism is particularly well-suited for datasets with high-dimensional channels, enhancing the model's capacity to learn complex spatiotemporal patterns and thereby improving overall prediction performance.

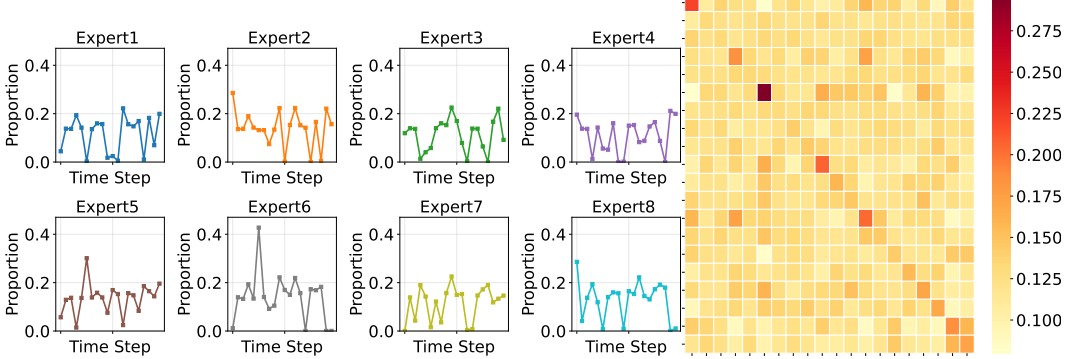

Figure 6: Visualization of multi-expert cross-channel dependencies.

**Bias Term Ablation.** Excessive decomposition and over-reliance on expert models can introduce redundant features, ultimately degrading overall performance. Moreover, both the MSRC's expert selection mechanism and the MEC's prediction process are susceptible to disruption by fine-grained patterns that are difficult to capture. These subtle patterns interfere with the intended inference behavior of individual experts, leading to suboptimal decisions. To address this issue, we explicitly model such elusive signals as compensatory bias terms, designed to counteract their adverse effects. By integrating these bias terms using an optimal configuration of decoupled layers and expert modules, we systematically evaluate their impact on model performance. As illustrated in Fig. 7, our approach effectively mitigates the detrimental influence of fine-grained noise, resulting in enhanced predictive accuracy.

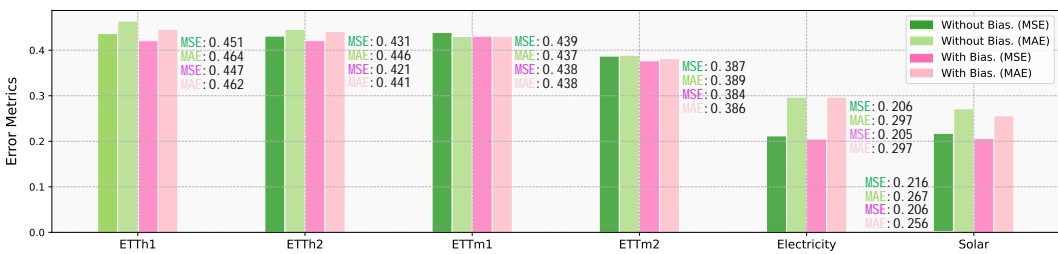

Figure 7: Visualization of the model performance with the addition of bias Terms

Table 2: The look-back window for the experiment is 96, and the forecasting horizons are {96, 192, 336, 720}. All results are averaged.

| | MSRC | | | | | | MEC | | | | | |
|---|---|---|---|---|---|---|---|---|---|---|---|---|
| Datasets | ETTh1 | | ETTm1 | | Weather | | ECL | | Solar | | PEMS04 | |
| Metrics | MSE | MAE | MSE | MAE | MSE | MAE | MSE | MAE | MSE | MAE | MSE | MAE |
| Linear | 0.432 | 0.437 | 0.379 | 0.400 | 0.241 | 0.269 | 0.174 | 0.229 | 0.197 | 0.254 | 0.102 | 0.212 |
| Convolution | 0.427 | 0.438 | 0.381 | 0.403 | 0.248 | 0.274 | 0.181 | 0.234 | 0.222 | 0.234 | 0.105 | 0.216 |
| Transformer | 0.458 | 0.464 | 0.374 | 0.397 | 0.262 | 0.301 | 0.192 | 0.246 | 0.256 | 0.271 | 0.129 | 0.232 |
| MLP | 0.420 | 0.433 | 0.371 | 0.395 | 0.241 | 0.271 | 0.165 | 0.259 | 0.194 | 0.254 | 0.099 | 0.210 |

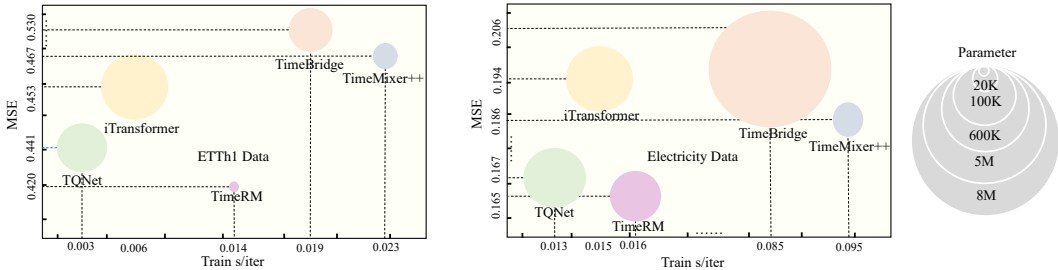

Figure 8: Visualization of the model efficiency

**Expert Model Analysis.** Expert models within the MSRC and MEC modules are crucial for analyzing temporal patterns and channel relationships. We selected commonly used methods in current research as expert models. As shown in Table 2, by selecting and comparing MLP layers Lin et al. (2024); Yi et al. (2024), linear layers Zeng et al. (2023), convolutional layers Wang et al. (2025a), and Transformer layers Liu et al. (2024b), we obtained expert models that are more suitable for TimeRM.

**Efficiency Analysis.** Fig. 8 shows a comparison of TimeRM with current SOTA (State-Of-The-Art) methods in terms of computational efficiency. In the bubble chart, for the ETTh1 dataset, we employed the MoE strategy for temporal pattern decomposition, resulting in slower computational speed compared to TQNet. However, our model achieves lower parameter count and lower error. For the Electricity dataset, TimeRM utilizes the MEC module, demonstrating significant overall performance advantages in parameter efficiency, computational speed, and error reduction.

## 5 CONCLUSION

This study tackles the key challenge in time series forecasting where limited prediction arises from residuals containing mixed fluctuations and patterns by proposing the TimeRM framework. It decomposes time series into periodic and residual components, employs a Multi-Level Decoupling module to extract sub-residuals with distinct patterns to supplement the main model, utilizes a Multi-Level Sub-Residual Compensation module to capture intertwined temporal patterns and mitigate fluctuation interference, and incorporates the Multi-Expert Coupling module to enhance expert models for complex channel features. Experiments show TimeRM outperforms state-of-the-art methods, improving forecasting accuracy and robustness.

## 6 ETHICS STATEMENT

This research focuses on improving the accuracy and effectiveness of time series forecasting through the development of a novel residual modeling framework, TimeRM. The work is conducted in accordance with ethical principles for scientific research, ensuring honesty, integrity, and transparency. All data used in the experiments are publicly available or synthetically generated for academic purposes, with no involvement of human subjects, personal privacy information, or sensitive data. The proposed method aims to advance the field of time series analysis and support downstream applications in areas such as finance, healthcare, and climate prediction. We declare no conflicts of interest, and the research is not funded by any organization that may benefit commercially from the outcomes in an unethical manner.

## 7 REPRODUCIBILITY STATEMENT

Our code has been submitted in the supplementary materials. For the proposed MSRC and MEC modules, specific parameter configurations are provided within the code, which is available in the scripts directory. All datasets used in the experiments are publicly accessible. Our experimental results are reproducible. The usage of the large language model is described in the Appendix C.

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

# A    DETAILS OF TIMERM

## A.1    MULTI-SUB-RESIDUAL

In an ideal time series decomposition framework, the residual component should capture purely random and unpredictable fluctuations, devoid of any systematic structure. However, simple average pooling often fails to fully extract complex long-term trends or evolving seasonal patterns, leading to underfitting. As a result, portions of the trend, cyclical, and other structured components may inadvertently persist in the residuals, compromising their interpretability and utility. To better isolate true stochastic noise, we propose iteratively extracting sub-residuals to refine the characterization of random fluctuations. This process are shown in Algorithm 3.

---

**Algorithm 3** Multi-Level Residual Decompose

---

1: **procedure** MULTI-LEVEL-DECOMPOSITION($x$)
2:     $n$                                                           ▷ Decomposition level
3:     $residuals \leftarrow [\ ]$
4:     **for** $i \leftarrow 1$ **to** $n$ **do**
5:         $residual \leftarrow H,$    where $H \in \mathbb{R}^{B \times L \times C}$
6:         $residual, stationarytrend \leftarrow$ SeriesDecomp($residual$)
7:         **append** $residual$ **to** $residuals$
8:     **end for**
9:     **return** $residuals$                                       ▷ Output
10: **end procedure**

---

Empirically, we observe that the residual series frequently contains not only white noise but also interpretable systematic patterns, including persistent long-term trends, event-induced abrupt shifts, and time-varying periodic behaviors. To address this contamination, we introduce a Multi-Sub-Residual Compensation (MSRC) strategy. MSRC identifies and models multiple underlying sub-residual structures prior to final residual analysis, quantifies their contributions to the primary residual, and compensates for them, thereby yielding a cleaner and more accurate residual representation that minimizes noise interference.

Furthermore, we design a Multi-Expert Coupling (MEC) mechanism that enables specialized experts to independently learn distinct types of non-periodic patterns, such as structural breaks, transient shocks, and trend deviations. These expert outputs are then dynamically integrated with the periodic forecast through adaptive weighting, allowing for effective calibration of the baseline prediction. This dual approach enhances the model's ability to capture complex temporal dynamics and significantly improves overall forecasting accuracy.

### A.2    REMOVE PERIODIC INFORMATION

Inspired by CycleNet Lin et al. (2024), we incorporate its proposed RCF module, which explicitly models and separates the periodic and residual components in time series data, thereby significantly enhancing the model's ability to capture complex temporal patterns. Specifically, given $C$ channels and a prior cycle length $L_c$, we introduce a learnable parameter matrix $Q \in \mathbb{R}^{L_c \times C}$, where each row corresponds to a specific phase within the periodic cycle. The index $\tau_t$ denotes the position within the cycle corresponding to time step $t$. Given an input sequence $x$, we determine the cyclic positions $\{\tau_t\}$ for each time step and retrieve the associated periodic features from $Q$. This process can be formulated as:

$$C_x = [Q^{(\tau_1)}, Q^{(\tau_7)}, \ldots, Q^{(\tau_{(L|L_c)})}, \ldots], \ \ C_y = [Q^{(\tau_1)}, Q^{(\tau_7)}, \ldots, Q^{(\tau_{(N|L_c)})}, \ldots]. \tag{9}$$

## B    EXPERIMENTS

### B.1    BASELINE

In our experiments, we benchmark against current state-of-the-art (SOTA) methods for comparative analysis. In time series forecasting, the outstanding performer TQNet Lin et al. (2025), TimeBridge Liu et al. (2024a), FLDManba Liu et al. (2024a), TimeMixer++ Wang et al. (2025b), TimeMixer Wang et al. (2024a), CycleNet Lin et al. (2024) and iTransformer Liu et al. (2024b)

### B.2    DATASETS

The characteristics of the dataset used in the experiment are shown in Table 3, and the specific details are as follows:

- The Electricity Transformer Temperature (ETT) Ren et al. (2024) dataset records the operational data from electricity transformers at two substations in China. The key challenge is to predict the oil temperature of the transformer, which is crucial for predictive maintenance and preventing failures. It is a canonical dataset for evaluating multivariate time series forecasting, where the model must leverage multiple related variables to predict the target temperature. The dataset is split into ETTh1, ETTh2, ETTm1 and ETTm2.

- The Weather dataset Wu et al. (2021) is derived from atmospheric reanalysis data, providing high-resolution meteorological measurements over North America. It includes multiple correlated variables such as temperature, pressure, humidity, and wind speed, forming a large-scale multivariate time series forecasting task.

- The Electricity Wu et al. (2021) dataset consists of the hourly electricity consumption of 321 customers. It captures complex temporal patterns including daily, weekly, and seasonal cycles, as well as holiday effects. Accurate electricity load forecasting is vital for grid stability and energy market operations.

- The Solar-Energy dataset Lai et al. (2018) contains solar power generation measurements from multiple photovoltaic sites in the United States, sampled at a 10-minute interval over a year. The output exhibits strong diurnal and seasonal patterns and is sensitive to environmental factors such as cloud cover and solar irradiance. Accurate forecasting supports renewable energy integration into the power grid.

- The PEMS dataset Liu et al. (2022a) comprises traffic flow data collected by sensors deployed on highways in California, USA. It represents a classic spatio-temporal forecasting problem, as the traffic flow at one sensor is influenced by the flow at upstream and downstream sensors. Capturing these spatial dependencies is as important as modeling temporal patterns.

Table 3: Detailed information about the datasets.

| Dataset | Timesteps | Interval | Channel | Domain |
|---------|-----------|----------|---------|--------|
| ETTh1 | 14400 | 1 hour | 7 | Electricity |
| ETTh2 | 14400 | 1 hour | 7 | Electricity |
| ETTm1 | 57600 | 15 mins | 7 | Electricity |
| ETTm2 | 57600 | 15 mins | 7 | Electricity |
| Weather | 52696 | 10 mins | 21 | Weather |
| Electricity | 26304 | 1 hour | 321 | Electricity |
| Solar | 52560 | 10 mins | 137 | Energy |
| PEMS04 | 16992 | 5 mins | 307 | Transportation |
| PEMS08 | 17856 | 5 mins | 170 | Transportation |

### B.3 RESULTS

The complete experimental results are summarized in Table 4. In terms of both MSE and MAE, the proposed TimeRM method ranks among the top two across all prediction horizons on average. Specifically, the MSRC module effectively captures intricate temporal patterns within residual components by deeply exploring their interactions along the time dimension, demonstrating superior modeling capability on low-dimensional datasets (ETTh1, ETTh2, ETTm1, ETTm2, and Weather). In contrast, the MEC module exhibits stronger predictive performance on high-dimensional datasets (Electricity, Solar, and PEMS). Overall, our method consistently achieves the best performance in terms of MSE, while attaining second-best results in most cases under MAE.

Notably, although the MAE performance is slightly inferior to some existing methods, the significant advantage in MSE indicates that our explicit modeling of residual components effectively suppresses large prediction errors caused by random fluctuations, thereby enhancing the stability and robustness of the forecasts. Given that current state-of-the-art methods differ in design objectives and optimization priorities, they naturally exhibit strengths on specific types of datasets. Considering the comprehensive evaluation across multiple metrics and datasets, the proposed approach achieves the best overall performance.

## C THE USE OF LLMS

This paper uses LLM to check for spelling errors and correct grammatical mistakes that may occur during the writing process.

Table 4: Long-term series forecasting results for a sequence length of 96. The forecasting horizons are set to {96, 192, 336, 720}. The best and the second-best results are highlighted in red and blue, respectively.

| Models | | TimeRM(MRSC) | | TimeRM(MEC) | | TQNet 2025 | | TimeBridge 2025 | | FLDMamba 2025 | | TimeMixer++ 2025 | | TimeMixer 2024 | | CycleNet 2024 | | iTransformer 2024 | |
|---|---|---|---|---|---|---|---|---|---|---|---|---|---|---|---|---|---|---|---|
| Datasets | Metrics | MSE | MAE | MSE | MAE | MSE | MAE | MSE | MAE | MSE | MAE | MSE | MAE | MSE | MAE | MSE | MAE | MSE | MAE |
| ETTh1 | 96 | 0.366 | 0.398 | 0.366 | 0.399 | 0.371 | 0.393 | 0.428 | 0.439 | 0.374 | 0.393 | 0.378 | 0.399 | 0.375 | 0.395 | 0.386 | 0.405 | 0.390 | 0.411 |
| | 192 | 0.415 | 0.425 | 0.422 | 0.429 | 0.428 | 0.426 | 0.478 | 0.476 | 0.427 | 0.422 | 0.440 | 0.431 | 0.436 | 0.428 | 0.441 | 0.436 | 0.443 | 0.442 |
| | 336 | 0.449 | 0.445 | 0.456 | 0.454 | 0.476 | 0.446 | 0.527 | 0.510 | 0.447 | 0.441 | 0.499 | 0.459 | 0.496 | 0.455 | 0.487 | 0.458 | 0.482 | 0.469 |
| | 720 | 0.447 | 0.462 | 0.457 | 0.459 | 0.487 | 0.470 | 0.686 | 0.604 | 0.469 | 0.463 | 0.549 | 0.510 | 0.520 | 0.484 | 0.503 | 0.491 | 0.496 | 0.488 |
| | Avg | 0.420 | 0.433 | 0.425 | 0.435 | 0.441 | 0.434 | 0.530 | 0.507 | 0.434 | 0.430 | 0.467 | 0.450 | 0.457 | 0.441 | 0.454 | 0.448 | 0.453 | 0.453 |
| ETTh2 | 96 | 0.287 | 0.340 | 0.295 | 0.348 | 0.295 | 0.343 | 0.296 | 0.339 | 0.287 | 0.337 | 0.287 | 0.340 | 0.298 | 0.344 | 0.297 | 0.349 | 0.329 | 0.371 |
| | 192 | 0.362 | 0.389 | 0.363 | 0.392 | 0.367 | 0.393 | 0.366 | 0.392 | 0.370 | 0.388 | 0.373 | 0.392 | 0.372 | 0.396 | 0.380 | 0.400 | 0.402 | 0.414 |
| | 336 | 0.407 | 0.424 | 0.405 | 0.425 | 0.417 | 0.427 | 0.410 | 0.427 | 0.412 | 0.425 | 0.423 | 0.434 | 0.431 | 0.439 | 0.428 | 0.432 | 0.440 | 0.445 |
| | 720 | 0.421 | 0.441 | 0.426 | 0.443 | 0.433 | 0.446 | 0.432 | 0.440 | 0.419 | 0.438 | 0.436 | 0.448 | 0.450 | 0.458 | 0.427 | 0.445 | 0.480 | 0.477 |
| | Avg | 0.370 | 0.399 | 0.372 | 0.402 | 0.378 | 0.402 | 0.376 | 0.399 | 0.372 | 0.396 | 0.380 | 0.404 | 0.388 | 0.409 | 0.383 | 0.407 | 0.413 | 0.427 |
| ETTm1 | 96 | 0.314 | 0.361 | 0.308 | 0.356 | 0.311 | 0.353 | 0.367 | 0.378 | 0.318 | 0.360 | 0.317 | 0.359 | 0.319 | 0.360 | 0.334 | 0.368 | 0.319 | 0.366 |
| | 192 | 0.349 | 0.381 | 0.350 | 0.380 | 0.356 | 0.378 | 0.419 | 0.416 | 0.365 | 0.384 | 0.368 | 0.387 | 0.360 | 0.381 | 0.377 | 0.391 | 0.377 | 0.397 |
| | 336 | 0.381 | 0.401 | 0.383 | 0.403 | 0.390 | 0.401 | 0.478 | 0.453 | 0.404 | 0.409 | 0.402 | 0.408 | 0.389 | 0.403 | 0.426 | 0.420 | 0.417 | 0.422 |
| | 720 | 0.438 | 0.438 | 0.447 | 0.441 | 0.452 | 0.440 | 0.543 | 0.504 | 0.464 | 0.441 | 0.450 | 0.441 | 0.447 | 0.441 | 0.491 | 0.459 | 0.487 | 0.463 |
| | Avg | 0.371 | 0.395 | 0.372 | 0.395 | 0.377 | 0.393 | 0.452 | 0.438 | 0.389 | 0.399 | 0.384 | 0.399 | 0.379 | 0.396 | 0.407 | 0.410 | 0.400 | 0.412 |
| ETTm2 | 96 | 0.167 | 0.250 | 0.169 | 0.252 | 0.173 | 0.256 | 0.175 | 0.254 | 0.173 | 0.253 | 0.175 | 0.256 | 0.170 | 0.253 | 0.180 | 0.264 | 0.182 | 0.266 |
| | 192 | 0.232 | 0.295 | 0.237 | 0.299 | 0.238 | 0.298 | 0.240 | 0.298 | 0.240 | 0.299 | 0.238 | 0.300 | 0.238 | 0.301 | 0.250 | 0.309 | 0.248 | 0.306 |
| | 336 | 0.292 | 0.330 | 0.292 | 0.333 | 0.301 | 0.340 | 0.302 | 0.338 | 0.301 | 0.337 | 0.298 | 0.339 | 0.294 | 0.335 | 0.311 | 0.348 | 0.312 | 0.346 |
| | 720 | 0.383 | 0.386 | 0.389 | 0.392 | 0.397 | 0.396 | 0.402 | 0.395 | 0.401 | 0.397 | 0.437 | 0.420 | 0.393 | 0.399 | 0.412 | 0.407 | 0.414 | 0.404 |
| | Avg | 0.269 | 0.315 | 0.272 | 0.319 | 0.277 | 0.323 | 0.280 | 0.321 | 0.279 | 0.322 | 0.287 | 0.329 | 0.274 | 0.322 | 0.288 | 0.332 | 0.289 | 0.330 |
| Weather | 96 | 0.154 | 0.201 | 0.167 | 0.211 | 0.157 | 0.200 | 0.172 | 0.207 | —— | —— | 0.162 | 0.210 | 0.158 | 0.203 | 0.174 | 0.214 | 0.163 | 0.212 |
| | 192 | 0.206 | 0.250 | 0.212 | 0.253 | 0.206 | 0.245 | 0.220 | 0.248 | —— | —— | 0.213 | 0.252 | 0.207 | 0.247 | 0.221 | 0.254 | 0.211 | 0.254 |
| | 336 | 0.263 | 0.291 | 0.266 | 0.291 | 0.262 | 0.287 | 0.276 | 0.290 | —— | —— | 0.273 | 0.294 | 0.262 | 0.289 | 0.278 | 0.296 | 0.273 | 0.299 |
| | 720 | 0.342 | 0.341 | 0.346 | 0.344 | 0.344 | 0.342 | 0.352 | 0.344 | —— | —— | 0.350 | 0.344 | 0.344 | 0.344 | 0.358 | 0.349 | 0.351 | 0.348 |
| | Avg | 0.241 | 0.271 | 0.248 | 0.275 | 0.242 | 0.269 | 0.255 | 0.272 | —— | —— | 0.250 | 0.275 | 0.243 | 0.271 | 0.258 | 0.278 | 0.249 | 0.278 |
| Electricity | 96 | 0.170 | 0.274 | 0.135 | 0.229 | 0.138 | 0.236 | 0.183 | 0.258 | 0.137 | 0.234 | 0.156 | 0.247 | 0.136 | 0.229 | 0.148 | 0.240 | 0.165 | 0.274 |
| | 192 | 0.176 | 0.276 | 0.151 | 0.245 | 0.154 | 0.247 | 0.188 | 0.266 | 0.158 | 0.251 | 0.171 | 0.254 | 0.152 | 0.244 | 0.162 | 0.253 | 0.185 | 0.292 |
| | 336 | 0.192 | 0.290 | 0.168 | 0.263 | 0.169 | 0.264 | 0.205 | 0.284 | 0.182 | 0.173 | 0.187 | 0.276 | 0.170 | 0.264 | 0.178 | 0.269 | 0.197 | 0.304 |
| | 720 | 0.238 | 0.329 | 0.205 | 0.297 | 0.206 | 0.300 | 0.248 | 0.318 | 0.200 | 0.292 | 0.228 | 0.312 | 0.212 | 0.299 | 0.225 | 0.317 | 0.231 | 0.332 |
| | Avg | 0.202 | 0.298 | 0.165 | 0.259 | 0.167 | 0.262 | 0.206 | 0.282 | 0.170 | 0.238 | 0.186 | 0.272 | 0.168 | 0.259 | 0.178 | 0.270 | 0.194 | 0.301 |
| Solar | 96 | 0.207 | 0.286 | 0.176 | 0.250 | 0.173 | 0.233 | 0.186 | 0.228 | 0.202 | 0.233 | 0.199 | 0.247 | 0.215 | 0.295 | 0.190 | 0.247 | 0.203 | 0.237 |
| | 192 | 0.216 | 0.289 | 0.192 | 0.253 | 0.199 | 0.257 | 0.218 | 0.250 | 0.230 | 0.254 | 0.230 | 0.271 | 0.236 | 0.301 | 0.210 | 0.266 | 0.233 | 0.261 |
| | 336 | 0.221 | 0.284 | 0.201 | 0.256 | 0.211 | 0.263 | 0.239 | 0.261 | 0.254 | 0.265 | 0.248 | 0.286 | 0.252 | 0.307 | 0.217 | 0.266 | 0.248 | 0.273 |
| | 720 | 0.215 | 0.278 | 0.207 | 0.256 | 0.209 | 0.270 | 0.245 | 0.277 | 0.252 | 0.271 | 0.251 | 0.287 | 0.244 | 0.305 | 0.223 | 0.266 | 0.249 | 0.275 |
| | Avg | 0.217 | 0.284 | 0.194 | 0.254 | 0.198 | 0.256 | 0.222 | 0.254 | 0.235 | 0.256 | 0.232 | 0.273 | 0.237 | 0.302 | 0.210 | 0.261 | 0.233 | 0.262 |
| PEMS04 | 96 | 0.144 | 0.260 | 0.075 | 0.180 | 0.083 | 0.195 | —— | —— | 0.075 | 0.182 | 0.089 | 0.199 | 0.079 | 0.183 | 0.076 | 0.182 | 0.077 | 0.180 |
| | 192 | 0.148 | 0.269 | 0.090 | 0.200 | 0.113 | 0.231 | —— | —— | 0.094 | 0.193 | 0.127 | 0.241 | 0.087 | 0.195 | 0.089 | 0.201 | 0.093 | 0.201 |
| | 336 | 0.203 | 0.210 | 0.105 | 0.218 | 0.142 | 0.261 | —— | —— | 0.105 | 0.217 | 0.172 | 0.281 | 0.110 | 0.214 | 0.136 | 0.247 | 0.125 | 0.236 |
| | 720 | 0.188 | 0.299 | 0.124 | 0.239 | 0.178 | 0.296 | —— | —— | 0.130 | 0.243 | 0.221 | 0.323 | 0.148 | 0.251 | 0.182 | 0.282 | 0.164 | 0.275 |
| | Avg | 0.171 | 0.260 | 0.099 | 0.209 | 0.129 | 0.246 | —— | —— | 0.099 | 0.209 | 0.152 | 0.261 | 0.105 | 0.259 | 0.118 | 0.226 | 0.113 | 0.222 |

