# OpenReview forum: "TimeRM: Multi-Expert Residual Modeling for Long-Term Time Series Forecasting"
_ICLR.cc/2026/Conference — ICLR 2026 Conference Withdrawn Submission_

### Official Review · Reviewer_RbzB · 2025-10-29

**Soundness:** 3
**Presentation:** 2
**Contribution:** 2
**Rating:** 2
**Confidence:** 3

**Summary:**

This paper proposes the TimeRM framework to model residual components in time-series forecasting: MLD extracts hierarchical sub-residuals; MSRC uses a mixture-of-experts to capture temporal patterns; MEC integrates high-dimensional channel features.

**Strengths:**

1. The paper includes experiments on multiple standard datasets (ETT, Weather, Electricity, Solar, PEMS) with thorough ablation studies.

2. The paper explores how complex temporal patterns within residuals affect forecasting, which is practically meaningful.

**Weaknesses:**

1. It would be better if the author could clearly articulate what is fundamentally novel beyond combining previous techniques. The paper mentions inspirations from CycleNet and Time-MoE, but the conceptual relationship and motivation for combining them are not sufficiently explained. The authors should compare their method with a simple combination of the original CycleNet and Time-MoE designs.

2. The method, with three components, is overly complicated. The reader might feel that the method is a stack of tricks and previous methods, diluting the core novel contribution of this method.

4. The claim in the abstract "outperforms current state-of-the-art methods" is not fully supported. Table 1 shows TimeRM is often second-best in MAE.

3. The presentation could be improved:
 - $z$ in Alg. 1 seems the same as $r$. Use $r$ directly (consistent with Eq. 3) to improve clarity.
 - is "Multi-Level Sub-Residual Compensate" the same with "Multi-Sub-Residuals Compensate"
 - Alg. 2 can be removed, which is just a model processing all the residuals.
 -  **bold** in Tab 1 can be misleading. I suggest the author use **bold**  for best result and $\underline{\text{underline}}$ for second best result, following previous methods.
 -  Line 672: The reference of FLDMamba links to Timebridge.

**Questions:**

1. What specific innovation does TimeRM contribute to residual modeling that hasn't been explored? (Weaknesses 1)

2. The bias terms are not a principled solution—what do they represent theoretically?

---

### Official Review · Reviewer_2DU4 · 2025-10-31

**Soundness:** 1
**Presentation:** 2
**Contribution:** 1
**Rating:** 2
**Confidence:** 4

**Summary:**

This paper addresses a key challenge in time series forecasting where complex residuals, containing unmodeled dynamics and entangled temporal patterns, limit predictive accuracy. The authors propose TimeRM, a residual modeling framework that first decomposes the time series into periodic and residual components. A Multi-Level Decoupling (MLD) module processes the residuals to extract sub-residuals, supplying the main model with previously neglected hierarchical dynamics. To capture entangled patterns, a Mixture-of-Experts (MoE) based module called MSRC compensates the primary residual, reducing the impact of unmodeled dynamics. For high-dimensional data, an alternative Multi-Expert Coupling (MEC) module is introduced to enhance the prediction of neglected channel features. Extensive experiments show that the TimeRM framework outperforms current state-of-the-art methods on long-term forecasting tasks.

**Strengths:**

1.Adaptive Design for Different Data Characteristics: The framework is highly adaptive, moving beyond a one-size-fits-all solution by providing two specialized modules for different data properties.

**Weaknesses:**

1.Fundamental Assumption on Residuals: The framework's core is built on the assumption that residuals contain valuable, predictable patterns. If the residuals of a time series are genuinely random white noise, then the MLD and MSRC/MEC modules are essentially modeling noise, which is highly likely to lead to overfitting and degraded performance.

2.Cost of MoE Architecture: The MSRC module adopts a Mixture-of-Experts (MoE) architecture. This inevitably leads to a significant increase in parameter count compared to a single model.

3.Robustness of Hyperparameters: The model introduces several critical hyperparameters, and its performance may not be robust.

4.Reliance on Priori Period Length: The model borrows from CycleNet in its preprocessing stage, using a learnable parameter matrix to remove periodicity.

**Questions:**

1.Regarding White Noise Residuals: How would TimeRM perform if applied to a synthetic dataset where the residuals are known to be purely random white noise? Would the model learn to ignore this noise, or would it overfit to it?

2.Regarding MoE Overhead: Could the authors provide a more detailed cost-benefit analysis of the MSRC module?

---

### Official Review · Reviewer_UCuq · 2025-11-01

**Soundness:** 2
**Presentation:** 3
**Contribution:** 2
**Rating:** 6
**Confidence:** 2

**Summary:**

The authors propose a framework named TimeRM to improve the time series forecasting by combining residual modeling with the mixture of expert architecture. The key idea is to decompose the residuals and capture the residual patterns using MOE. The authors evaluate their proposed method against baseline models and show that it achieves superior performance.

**Strengths:**

1. Overall this paper is clearly written and relatively easy to understand.

2. The proposed method is moderately novel and may be of interest to researchers in the field.

3. The authors show familiarity with the recent literature in time series forecasting.

**Weaknesses:**

1. In addition to MAE and MSE, the authors should evaluate their proposed method with MAPE (mean absolute percentage error) which is robust under different scales of the time series values.

2. The authors should also evaluate their proposed method on standard benchmark datasets for time series forecasting, such as the M4 competition dataset.

3. The authors should fix the various format errors and typos in their Related Works section, such as “Bandara et al. Bandara et al. (2025)” and “Wu et al. Zhang et al. (2025b)”.

**Questions:**

1. How does the model performance change with C, the number of variates?

2. Should we use more experts when the value of C is larger?

3. Why is residual modeling necessary? Can we apply the TimeRM framework to the original time series instead of their residuals?

---

### Official Review · Reviewer_UAWU · 2025-11-01

**Soundness:** 2
**Presentation:** 3
**Contribution:** 1
**Rating:** 0
**Confidence:** 5

**Summary:**

This paper proposes a residual modeling method called TimeRM, which decomposes a time series into its seasonal and residual components and models the residual part using a designed module to predict future sequences.

**Strengths:**

S1: The paper, like previous methods, focuses on residuals as the main entry point for research, and the rationale is sound.

S2: The presentation of each module in this paper is clear.

**Weaknesses:**

W1: This paper lacks originality. Most of the core components are standard or well-established modules from existing literature, with little novel contribution.

W2: The paper’s review of SOTA methods is not comprehensive. For example, it overlooks Linear-based methods such as FITS [1], RNN-based improvements like PGN [2], and Attention-based approaches such as Leddam [3].

[1] FITS: Modeling Time Series with 10k Parameters. In The Twelfth International Conference on Learning Representations.

[2] PGN: The RNN's New Successor is Effective for Long-Range Time Series Forecasting. In The Thirty-eighth Annual Conference on Neural Information Processing Systems.

[3] Revitalizing Multivariate Time Series Forecasting: Learnable Decomposition with Inter-series Dependencies and Intra-series Variations Modeling. In Proceedings of the 41st International Conference on Machine Learning.

W3: The experiments lack critical details. The paper does not provide any information about the hyperparameter search space, making it unclear whether the authors performed hyperparameter tuning for all baseline methods on the validation set before reporting results on the test set. This process needs to be clearly described:

(a) If hyperparameter search was conducted, the full search space for each dataset and task, as well as the final selected parameters, should be provided.

(b) If no such search was performed, the experimental results are questionable. Even with the same parameters and random seeds, results can vary across different hardware platforms. To ensure fair and rigorous comparisons, all methods should be run on the same platform, using a unified and sufficiently broad hyperparameter search space, selecting the best parameters on the validation set, and then evaluating on the test set. This is necessary to eliminate the influence of platform differences and hyperparameter sensitivity on model performance, and is essential for fair model comparison. If this was not done, additional experiments are required.

(c) Regarding model efficiency analysis, it should be clarified whether the compared baseline models used consistent hidden dimensions, batch sizes, number of layers, etc., to ensure fairness in efficiency comparisons.

W4: The model lacks consistency and generalization ability. Currently, TimeRM primarily uses MSRC on low-dimensional datasets and MEC on high-dimensional datasets. This indicates that different processing modules must be selected for different datasets, suggesting that TimeRM’s two implementations are scenario-dependent and lack generalization capability.

W5: The effectiveness of the main contribution to time series prediction is questionable. The final prediction process in the paper consists of two steps: aligning and integrating the residual analysis with the seasonal information Cy, resulting in the final output y. Is it possible that most of the observable performance gains of the model primarily come from the fitting of Cy, by the seasonal branch/RCF module and the "addition of seasonality," rather than from the core contributions claimed in the paper, the multi-level residual decoupling (MLD) and multi-expert compensation (MSRC/MEC) design? This requires further experimental validation to confirm.

**Questions:**

See Weaknesses.

---

### Note · Authors · 2025-11-13

I have read and agree with the venue's withdrawal policy on behalf of myself and my co-authors.